# Shake It Up Baby Now: The Changing Focus on TWIST1 and Epithelial to Mesenchymal Transition in Cancer and Other Diseases

**DOI:** 10.3390/ijms242417539

**Published:** 2023-12-16

**Authors:** Dureali Mirjat, Muhammad Kashif, Cai M. Roberts

**Affiliations:** 1Arizona College of Osteopathic Medicine, Midwestern University, Glendale, AZ 85308, USA; 2Department of Pharmacology, Midwestern University, Downers Grove, IL 60515, USA

**Keywords:** TWIST1, epithelial to mesenchymal transition, drug resistance, cancer stem cells, novel therapies

## Abstract

TWIST1 is a transcription factor that is necessary for healthy neural crest migration, mesoderm development, and gastrulation. It functions as a key regulator of epithelial-to-mesenchymal transition (EMT), a process by which cells lose their polarity and gain the ability to migrate. EMT is often reactivated in cancers, where it is strongly associated with tumor cell invasion and metastasis. Early work on TWIST1 in adult tissues focused on its transcriptional targets and how EMT gave rise to metastatic cells. In recent years, the roles of TWIST1 and other EMT factors in cancer have expanded greatly as our understanding of tumor progression has advanced. TWIST1 and related factors are frequently tied to cancer cell stemness and changes in therapeutic responses and thus are now being viewed as attractive therapeutic targets. In this review, we highlight non-metastatic roles for TWIST1 and related EMT factors in cancer and other disorders, discuss recent findings in the areas of therapeutic resistance and stemness in cancer, and comment on the potential to target EMT for therapy. Further research into EMT will inform novel treatment combinations and strategies for advanced cancers and other diseases.

## 1. Introduction

Epithelial–mesenchymal transition (EMT) is a pathway through which cuboidal, non-motile epithelial cells transform and acquire mesenchymal cell properties [1]. Loss of cell adhesion, apical-basal polarity, and more prolonged and augmented cell migration are all characteristics of EMT [2]. The process of EMT was first studied in the context of embryogenesis and fetal development. The idea of such a transition dates back to the nineteenth century [3], though more extensive studies of EMT in development would not begin until some sixty years thereafter. As reviewed previously by Lim and Thiery, multiple distinct rounds of EMT take place from the time of implantation through organ formation [4]. It is also in the context of development that the gene *TWIST1* was first studied. *TWIST1* encodes a member of the essential basic helix–loop–helix family of transcription factors, and together with its binding partners, recognizes conserved E-box sequences in target promoters [5]. Simpson described in 1983 the phenotype of the loss of *twi*, the drosophila homolog of the human *TWIST1* gene, namely a lack of mesoderm and thus internal organs in developing flies [6]. In the same paper, Simpson observes a similar phenotype for the loss of *sna*, the homolog of *SNAIL*. Later work in mice led to the cloning of the mammalian homologs of *twi*, which were linked to bone development [7,8]. Work in mice demonstrated a vital role for *twist* in neural tube closure, the formation of mature tissues from the neural crest, and the formation of bone elsewhere [9,10,11,12]. In humans, mutations in *TWIST1* can lead to Saethre–Chotzen Syndrome, a disorder characterized by craniosynostosis (premature skull fusion), a fusion of or abnormal numbers of the fingers or toes, and in some cases, impairment of hearing or learning [13,14,15].

EMT is also essential in developing cancer and its eventual spread [16]. Bates used integrin αvβ6, Domínguez et al. used a new Snail1 antibody, and Brabletz et al. used nuclear β-catenin, and all concluded that EMT markers were found in invasive tumors [17,18,19]. Parallels were observed between fetal development and the progression of tumors. In the early stages of embryogenesis, TWIST1 controls EMT, cell migration, and tissue reorganization [20,21]. In the same way, increased cell motility and EMT are linked to TWIST1 expression in tumor cells, which suggests that TWIST1 may play a part in metastasis [22,23]. Post-EMT cells gain the ability to move through and migrate across the extracellular matrix, which is needed for these cells to invade and relocate to faraway locations [24,25]. EMT also provides immunological monitoring and resistance against anoikis.

Tumor cells re-express EMT genes, such as *TWIST1*, to jumpstart metastasis in response to a variety of signals [26]. One mechanism involves the direct upregulation of *TWIST1* transcription by STAT3 in a feedforward loop also involving interleukin-6 and Notch1 [27,28,29,30,31]. Hypoxic conditions as a result of rapid tumor growth and insufficient vasculature can also lead to TWIST1 expression via HIF-1α [32]. TWIST1+ motile cells are capable of leaving the hypoxic environment and may also lead to the degradation of the extracellular matrix since TWIST1 drives the upregulation of matrix metallopeptidases (MMPs) via IL-8 and NF-κB [33]. TWIST1 may also relieve hypoxia via neovascularization, as described in multiple cancer types [34,35,36]. Oxidative damage has also been linked to EMT induction by asbestos during the development of mesothelioma [37]. EMT is also commonly induced by TGFβ; Fan et al. found that TWIST1 manages the TGF-β/Smad3 signaling pathway, which helps cervical cancer grow and begins EMT [38].

As a result of the link between TWIST1 expression and metastatic spread, a large body of work exists that thoroughly describes EMT in this context. An excellent overview was compiled previously by Thiery [39]. While TWIST1 is considered by some to be a “master regulator” of EMT [26], it is far from the only marker of mesenchymal phenotype. Abnormal expression of E-cadherin is linked to a higher amount of TWIST1, as reviewed by Nawshad [40]. Studies also found that TWIST1 decreases the transcription of E-cadherin and α, β, and γ-catenins in breast cancer cells [41,42]. TWIST1 activity also leads to the enhanced production of matrix metallopeptidases MMP2 and MMP9 [33]. The hallmarks of cancer EMT also include this loss of E-cadherin and the upregulation of N-cadherin expression, referred to as a cadherin switch [43,44]. EMT can be reversed upon seeding of secondary tumor sites, a process termed mesenchymal–epithelial transition, or MET. Some markers that are lost when cells change back are Zeb1, Zeb2, SNAIL, vimentin (Vim-1), and N-cadherin [26,45,46]. Others are gained, such as desmoplakins, E-cadherin, plakophilin, occludin, claudin, and cytokeratin.

Further studies are needed to clarify what fundamental regulatory processes are involved in EMT and its many roles. While much work has been performed on the molecular activity of TWIST1 in the stimulation of cell migration, in recent years focus has begun to shift. Of particular interest are the roles of EMT outside the contexts of development and cancer metastasis (Figure 1). The remainder of this review describes new findings on TWIST1 and EMT. First, we will discuss how TWIST1 and EMT contribute to non-cancerous disease states. Next, we will summarize their roles in stemness and drug resistance in cancer. Finally, we will describe efforts to drug EMT for therapy and comment on future opportunities in the treatment of cancer and other diseases.

## 2. Beyond Cancers: Additional Pathologies Associated with TWIST1

TWIST1 has been implicated in diseases in all stages of life, including during pregnancy. As mentioned earlier, TWIST1 mutations can lead to developmental disorders, such as Saethre–Chotzen Syndrome, in the developing fetus. Conversely, TWIST1 levels in placental tissue can be used to identify a very different kind of gynecologic disorder. Jahanbin et al. found that quantifying TWIST1 expression can distinguish complete from partial hydatidiform moles in cases of molar pregnancy [47]. Specificity ranged from 60% in syncytiotrophoblasts to 75% in stromal cells, with the trend of TWIST1 expression being the opposite in the two cell types. This is in contrast to a prior study of TWIST1 for this same purpose, which claimed 100% specificity [48].

The association of TWIST1 with stemness and the fibroblast phenotype even in select adult tissues has led to its implication in diseases outside of the context of either development or cancer, particularly fibrosis. Iwano et al. first introduced the idea that fibrosis involves EMT, and Xue et al. only later expanded on their study to account for metastatic tumor cell development [49,50]. TWIST1 has also been linked to EMT in a model of one-sided ureteral obstruction that causes renal fibrosis [51]. Over time, TWIST1 expression rose in the tubular epithelia of the dilated tubules and in the larger interstitial areas where cells were often growing. In addition, a recent study of idiopathic pulmonary fibrosis found that TWIST1 protein expression and open chromatin at TWIST1 binding sites were both elevated in myofibroblasts. Furthermore, the overexpression of TWIST1 in these cells led to increased collagen I production [52].

Several additional disease states have been tied to EMT. For instance, TWIST1 gave rise to a steroid-resistant phenotype in a study of ulcerative colitis, in this case being expressed at high levels in neutrophils [53]. *TWIST1* was also linked to immune cells in patients with mastocytosis, where it was differentially methylated compared to healthy controls [54]. Neural diseases have also been associated with EMT or a related process, endothelial-to-mesenchymal transition (EndMT). A 2023 study used a rat model to establish a link between TWIST1 and schizophrenia. Loss of the microRNA miR-25-3p led to an elevated expression of TWIST and activation of the PI3K/Akt/GSK3β pathway across multiple brain regions [55]. In a study of cavernous malformations in neural tissue, immune activation and tissue reconstruction were linked to EndMT. EndMT within a minority cell population gave rise to fibroblast-like smooth muscle cells via TWIST1 [56]. Interestingly, EndMT can also be induced in liver sinusoidal endothelium by exposure to E. coli in a mouse model of nonalcoholic fatty liver disease [57]. In this case, EndMT was mediated by TWIST1 and the NFκB factor p65, which we and colleagues previously showed could together regulate IL-8 and matrix metallopeptidase production in cancer [33,58].

## 3. TWIST1, EMT, and Cancer Stem Cell Phenotypes

Many of the above disorders are characterized by abnormal differentiation of cells. Cancer is also characterized by heterogeneity and dysregulation of cell identity. Many tumors lose the properties of their tissue of origin, and tumor cells that undergo EMT may also acquire stem-like qualities [59,60]. Stemness refers to the capability of cells to divide unevenly, which lets them be both a source of newly differentiated cells and a storehouse of stem cell identity [61,62,63]. Dongre and Weinberg provide a review of EMT molecular mechanisms, including connections to stem cell phenotypes [64]. For instance, Yang et al. found that Bmi1, a regulator of self-renewal, was integral to TWIST1-dependent EMT [65]. Snail/Slug, TWIST1, and Zeb1/2 are classical EMT-TFs that bestow cancer stem cell features [61,66,67]. For example, the ratio of Snail to the epithelial marker E-cadherin correlated with stemness, drug resistance, and migration in ovarian cancer [68]. As reviewed by Garg in 2015, TWIST1 increases the production of VEGF, fibronectin, vimentin, beta-smooth muscle actin, and N-cadherin in stem-like cells [69]. Morel et al. used a model of how a breast tumor changes and grows over time to show that EMT controls cells’ abilities to become stem cells and grow into tumors [70]. This study demonstrated that TGF-β induced EMT and stemness, via Ras/MAPK, and extended prior work characterizing normal stem cells in the breast niche [71] to the context of cancer. In another breast cancer study, Choi et al. established that BMP-4 increases the expression of EMT biomarkers, including Slug, fibronectin, laminin, and N-cadherin. Also, BMP-4 enhances the sphere-forming capability of the mammary epithelial cell line MCF-10A and activates Notch signaling in these cells. They argue that BMP-4 boosts the expression of cancer stem cell markers, like CD44 and Nanog, in MDA-MB-231 cells [72]. These findings suggest that BMP-4 induces Smad4-dependent activation of Notch signaling in breast cancer cells, promoting EMT and stem cell properties. According to Hollier et al., the transcription factor FOXC2 is one of the key factors influencing the mesenchymal and stem cell characteristics amongst EMT- and stem cell-enriched breast cancer cell lines. FOXC2 is expressed in reaction to various EMT signaling pathways and is elevated in stem cell populations. Excess expression of Twist, Snail, TGFβ, or FOXC2 in breast cancer results in cells that are more mesenchymal and are characterized by cancer stem cell (CSC) markers, i.e., CD44+ and CD24-low [73]. The study concluded that FOXC2 or its related gene expression program could be an appropriate target for anti-EMT therapeutics for treating tumors that are EMT- or CSC-enriched, claudin-low, or basal B.

Several studies have focused on EMT-CSC links in pancreatic ductal adenocarcinoma (PDAC). As reviewed by Zhou et al., pancreatic CSC cell surface markers CD24 and CD44 encourage cell interactions, and c-Met reacts to ligands produced by active developmental pathways, like Notch and β-catenin, in pancreatic CSCs. These pathways stimulate genes that regulate stem cell characteristics [74]. According to research by Rhim et al., metastatic tumor cells had stem cell traits and a mesenchymal phenotype. They used a sensitive technique to tag and observe pancreatic epithelial cells in an animal pancreatic carcinoma model. This would support the idea that cancer stem cells are moving around in the blood, making the case for a more vital link between the stemness phenotype and the EMT program [75] in PDAC. Furthermore, metastatic pancreatic cancer cells with high levels of expression for the CSC markers nestin, ABCG2, and ALDH1A1 showed features of EMT, such as a reduced expression of E-cadherin. Both nestin and ABCG2 expression levels were also observed in the metastatic lesions of PDAC patients. Moreover, the ability of PDAC cells to form spheres and produce hepatic metastases was significantly reduced when nestin was silenced using shRNA [76]. The expression of nestin is, therefore, necessary for the metastatic capacity of human PDAC cells, which is correlated with both CSCs and EMT features. According to Wang et al., the self-renewal process of pancreatic CSCs has been associated with hedgehog (Hh) signaling. SMO is essential for controlling the characteristics of CSCs and preserving their capability to self-renew. This study demonstrates that SMO may control the EMT process in human pancreatic CSCs, affecting their ability to invade and migrate [77]. According to these findings, one efficient way to stop pancreatic CSCs from proliferating, invading, developing resistance, and spreading to the lungs is to block Hh signaling through SMO knockdown. Therefore, pancreatic cancer patients may benefit from targeted therapy inhibiting Hh signaling.

EMT and CSCs are linked across many additional cancers. In prostate cancer, TMPRSS4 expression was associated with resistance to anoikis and a CSC-EMT expression profile, including Bmi1, CD133, TWIST1, SOX2, and Slug [78]. According to Wang et al., prostate cancer cells’ ability to invade, migrate, and undergo EMT was improved by N-cadherin upregulation, whereas downregulation suppressed these processes. In turn, the overexpression of N-cadherin enhanced the expression of stemness factors, such as Klf4, Oct4, c-Myc, and Sox2, and enhanced tumor spheroid formation [79]. These results show that N-cadherin activates the ErbB signaling pathway to regulate the EMT and stemness of prostate cancer cells, indicating a central role of the N-cadherin/ErbB axis in the progression of prostate cancer metastasis. Meanwhile, Chen et al. recently found that USP51 expression in non-small cell lung carcinoma (NSCLC) positively correlated with the expression of CD44, SOX2, NANOG, and OCT4 stemness markers. When USP51 was depleted, NSCLC cell stemness and the expression of stemness markers declined. TWIST1 protein was more stable when USP51 was expressed ectopically because it reduced polyubiquitination. Moreover, TWIST1 could rescue stemness following USP51 loss [80]. Overall, this study shows that USP51 deubiquitinates TWIST1 to maintain NSCLC cells’ stemness. Its knockdown results in a reduction in both NSCLC cell growth and stemness, which could inform a path to novel treatment. In addition, when cirrhotic liver tissue is present, TGF-β encourages the progression of tumor-initiating cells [81]. The expression of stem markers is also modified in hepatic carcinoma cells due to TGF-β-induced EMT [81,82]. Furthermore, CSC markers were closely linked to EMT in esophageal cancer [83].

The link between TWIST1 and CSC phenotypes is not limited to solid tumors. Wang et al. demonstrated that TWIST1 elevated expression in the leukemia stem cells (LSCs) of patients with MLL-AF9+ acute myeloid leukemia (AML) and that KDM4C positively regulates its increased expression in an H3K9me3 demethylation-dependent manner. They also show that TWIST1 facilitates the start and continuation of MLL-AF9-mediated AML and is necessary for the viability, dormancy, and regeneration abilities of LSCs. Furthermore, TWIST1 directly interacts with and works with HOXA9 to induce AML in mice. Mechanistically, TWIST1 causes AML by activating the WNT5a/RAC1 axis [84]. This work reveals the crucial involvement of TWIST1 in LSC function and offers new mechanistic perspectives on the etiology of MLL-AF9+ AML. Overall, these studies have cemented stemness as an aspect of EMT that requires continued study.

## 4. TWIST1, EMT, and Drug Response in Cancer

### 4.1. Overview

TWIST1 and EMT can have a great impact on the response of cancer cells to a wide variety of drugs. In many cancers, tumor progression and metastasis correlate with drug resistance, and research on EMT has largely supported this trend. However, recent studies have demonstrated that this is not always the case. We and others have shown sensitization of select cells to therapy following EMT. In this section, we discuss the broad range of therapies whose efficacy may be impacted, either positively or negatively, by TWIST1 and other EMT factors. A summary of these agents and their applications is presented in Table 1.

### 4.2. EMT and Cytotoxic Drugs

Perhaps the largest body of work focuses on the impact of TWIST1 and other EMT factors in the response to traditional cytotoxic chemotherapy. We have published previously on TWIST1 and cisplatin in ovarian cancer. We found that TWIST1 acted via GAS6 and L1CAM to activate Akt in response to platinum exposure [85]. Aptecar et al. recently found that PTPN13 downregulation in ovarian cancer was linked to resistance to platinum and taxane therapy and EMT, as indicated by Snail, Slug, Zeb1, and Zeb2 expression [86]. Oxaliplatin and 5-fluorouracil resistance in colorectal cancer (CRC) can also result from the upregulation of Akt signals in tandem with STAT3 and TWIST1 activation. Deng et al. found that miR-15-5p reduced SIRT4 expression, leading to the growth and invasion of CRC cells. This phenotype was mediated in part by TWIST1 and Akt [87]. MicroRNA was also implicated in TWIST1-related changes in drug response in mesothelioma. The chemosensitizing agent butein activated miR-186-5p, reduced TWIST1 levels, and reduced cell survival and sphere-forming ability following cisplatin treatment [88]. miR-186 was similarly linked to a reduction in TWIST1 and a reversal of platinum resistance in ovarian cancer [89]. In the same vein, Parashar et al. developed a unique endometrioid ovarian cancer cell line model capable of forming spheroids in culture due to increased Akt/ERK signaling. This model demonstrated that cells growing as spheres had increased levels of several EMT markers, including TWIST1, Snail, vimentin, and MMPs. PI3K/Akt was responsible for both cisplatin resistance and EMT, as LY294002 treatment reversed EMT and sensitized cells to platinum [90]. Paclitaxel resistance in breast cancer cells was mediated by TWIST1 via the upregulation of Akt2 [91]. Each of these studies describes cross-talk between cell survival and proliferation signaling and the induction of chemoresistance and implicates EMT as a part of this process.

Many additional pathways can mediate EMT effects on chemoresponse. In ovarian cancer, lysophosphatidic acid (LPA) in the tumor microenvironment leads to EMT and tumor growth and invasion. Akt, JAK/STAT, and hedgehog signals are all implicated in this interaction [92]. Moreover, the inhibition of hedgehog signaling or treatment with resveratrol reversed LPA’s effects, induced autophagy, and sensitized cells to platinum-based therapy. This is in agreement with a former study by our colleagues, who found that the resveratrol analog pterostilbene could regulate STAT3 signaling and synergize with cisplatin in ovarian cancer [93]. In CRC cells, TWIST1 induced oxaliplatin resistance via the upregulation of microfibrillar-associated protein 2 (MFAP2) [94]. Chen et al. found that the Ras-related protein Rab31 induced TWIST1 and EMT in stomach cancer, and this, in turn, led to cisplatin resistance [95]. The same group has performed extensive work on the role of the translation initiation factor eIF5A2 in the context of EMT and drug response across cancers. In 2013, they showed that the inhibition of eIF5A2 by knockdown or N1-guanyl-1,7-diaminoheptane treatment prevented EMT and downstream doxorubicin resistance in hepatocellular carcinoma (HCC) [96]. A follow-up study in 2018 revealed that eIF5A2 conferred cisplatin resistance to gastric cancer cells via EMT induction. This phenotype was dependent on TWIST1, as TWIST1 knockdown prevented the effect of eIF5A2 on drug response [97]. eIF5A2 also conferred cisplatin resistance in a model of NSCLC, and while this study did not examine EMT directly, it found a correlation between eIF5A2 expression, autophagy, and hypoxia, a potential EMT stimulus as described earlier [32,98]. Hypoxia also has a complicated relationship with cancer-associated fibroblasts (CAFs). Nushtaeva et al. found that pulsed hypoxia stimulated CAFs to undergo an incomplete MET, though mesenchymal markers, such as TWIST1, were induced as an early response [99]. The presence of CAF-derived cells that had undergone this process worsened breast cancer tumor progression in vivo. A CAF gene signature was also associated with EMT and poor responses to neoadjuvant chemotherapy in a clinical study of gastric cancer [100]. Finally, TGFβ-expressing CAFs were associated with a chemoresistant TWIST1+ signature in hepatoblastoma. Interestingly, this signature was associated with an immunosuppressive tumor microenvironment and higher levels of aflatoxin–albumin in circulation [101].

### 4.3. Regulation of EMT Protein Expression

A number of studies have focused on the regulation of TWIST1 levels in the context of drug response. We and our colleagues recently showed that TWIST1 levels are regulated by the ubiquitin proteasome system and that phosphorylation by PKCα at key residues of TWIST1 could block ubiquitination [102]. Once its protein expression is stabilized, it is free to exert its pro-EMT and chemoresistant effects. Guan et al. showed that in triple-negative breast cancer, CDK1 leads to the de-ubiquitination of TWIST1 by USP29, which conferred resistance to both cisplatin and paclitaxel [103]. In CRC, TWIST1 is also protected from ubiquitination by interaction with hexokinase-2 (HK2). HK2 expression correlated with metastasis and resulted in oxaliplatin resistance [104]. A loss of HK2 could be overcome by TWIST1 overexpression, restoring the drug-resistant phenotype. Meanwhile, Li et al. also found that in myelodysplastic syndrome and AML, TWIST1 ubiquitination is blocked by O-GlcNAcylation. Furthermore, TWIST1 could upregulate O-GlcNAc transferase (OGT) expression, creating a feedforward loop that resulted in resistance to decitabine [105]. There may be other drug targets that could alter the post-translational modification and stability of TWIST1; Zhong et al. describe several proteins involved in this process [106]. Other EMT factors can be regulated in a similar fashion. In their study of FHL3 effects on oxaliplatin response, Cao et al. noted that FHL3 prevented the ubiquitination and turnover of Slug (Snail2) in gastric cancer cells [107]. Also, the phosphorylation of Snail regulates its subcellular location and function [18].

### 4.4. EMT and Multi-Drug Resistance

TWIST1 and EMT may act in such a manner that they affect responses to multiple drugs. Cross-resistance in cancer is often due to the expression of the multi-drug resistance protein MDR1, encoded by the gene ABCB1. Lai et al. showed that in HCC, NEIL3 bound to TWIST1 and upregulated the expression of ABCB1 [108]. This was dependent on TWIST1 DNA binding; TWIST1 has a central domain that binds to DNA at the consensus E-box sequence, which is found in the ABCB1 promoter. In CRC cells cross-resistant to 5-fluorouracil and oxaliplatin, the upregulation of ABCB1 and cellular invasiveness correlated with the expression of a host of EMT genes, including TWIST1 [109]. Cao et al. showed that oxaliplatin resistance in gastric cancer was mediated by MDR1 as a result of FHL3 expression in metastatic cells [107]. As its name implies, MDR1 can confer resistance to more than just cytotoxic drugs. For example, Solanes-Casado et al. examined colorectal cancer lines with acquired resistance to BI2536, an inhibitor of polo-like kinase 1 (PLK1). They found that resistant cells exhibited an increased expression of TWIST1 and vimentin, a loss of E-cadherin, and an induction of MDR1 expression [110]. Interestingly, the anti-cholesterol drug simvastatin was able to reverse this resistance.

### 4.5. EMT and Response to Anti-Angiogenic Therapy

PLK1 inhibitors are but one example of targeted therapies that have been developed that exploit mutations, phenotypes, or cytokine addictions unique to particular tumors. However, even these therapies are subject to the development of resistance, and EMT once again is often involved. One avenue of therapy is the inhibition of angiogenesis. As described above, tumors rely on vasculature to fuel their rapid growth, and can also rely on intravasation into the bloodstream for metastasis. Therefore, blocking the formation of vessels is a promising therapeutic strategy. However, cells can circumvent this blockade. One method of performing this is vascular mimicry, in which tumor cells form vessel-like structures substitute actual blood vessels [111]. Cheng et al. found that vascular mimicry by renal cancer cells led to resistance to the angiogenesis inhibitor pazopanib. The induction of this process was dependent on the long non-coding RNA (lncRNA) IGFL2-AS1 stabilizing androgen receptor (AR) expression, upon which TWIST1 could act downstream of AR [112]. A similar study demonstrated that another lncRNA, TANAR, protected TWIST1 itself from degradation, leading to AR-driven vascular mimicry [113]. In gastric cancer, resistance to the vascular endothelial growth factor receptor 2 inhibitor apatinib is mediated by the expression of COL1A2, which, in turn, was regulated at the transcriptional level by EP300 and TWIST1 [114]. The knockdown of any of these factors was sufficient to sensitize cells to apatinib once more.

### 4.6. EMT and Response to Growth Factor Inhibition

Another common target for newer drugs is signaling through the epidermal growth factor receptor (EGFR). Several studies have examined resistance to this approach in NSCLC. Hu et al. compared NSCLC cells sensitive and resistant to the EGFR inhibitor erlotinib and found that resistant cells exhibited increased Akt activity and an EMT phenotype, including a switch from TWIST2 to TWIST1 expression [115]. The same study found that the expression of LMNA reversed EMT via the inhibition of MAPK signals and preserved drug sensitivity. Peng et al. derived a gefitinib-resistant cell line and found that compared to the parent, the resistant line exhibited mesenchymal markers, such as TWIST1, vimentin, and Slug, as well as increased PI3K/Akt activity [116]. These findings echo the involvement of Akt survival signaling in drug resistance discussed at length for the cytotoxic drugs above. A study by Yu et al. also implicated TWIST1 in resistance to gefitinib, in this case via the non-coding RNAs Circ_0001658 and miR-409-3p [117].

### 4.7. EMT and Ferroptosis

Another novel idea in cancer therapy is the induction of a recently discovered cell-death pathway, ferroptosis. Wang et al. showed that erastin, a ferroptosis inducer, could kill gastric cancer cells, but that cells overexpressing cytoplasmic polyadenylation element binding protein 1 (CPEB1) were more sensitive than those with CPEB1 knocked down. TWIST1 was responsible for resistance to ferroptosis, and its expression was downregulated by CPEB1 [118].

### 4.8. EMT and Drug Sensitivity

Despite the many cases described here in which EMT or its constituent transcription factors are associated with drug resistance, there are some cases in which EMT is a marker of therapeutic vulnerability. For instance, we recently described a reduction in homologous recombination repair protein expression in multiple models of ovarian cancer EMT that sensitized the cells involved to treatment with cisplatin, the PARP inhibitor olaparib, and the DNA-PK inhibitor Nu-7441 [119]. This was based on prior work in which chemoresistant ovarian cancer CSCs were shown to be epithelial, while fast-dividing mesenchymal-like progeny was sensitive [120]. This also parallels work in breast cancer showing that TGF-β regulated the expression of a different set of DNA repair factors and resistance to another PARP inhibitor, ABT-888 [121]. In another example, Reisenauer et al. found that breast cancer cells that had undergone EMT were more sensitive to ophiobolin A, an agent derived from fungi, than cells that had not [122]. Furthermore, in a recent clinical trial of an MDM2 inhibitor, milademetan, for intimal sarcoma in Japan, TWIST1 expression was associated with favorable responses [123].

### 4.9. Intersection of CSC and Drug-Resistant Phenotypes

In many cases, stemness and drug resistance go hand in hand. In colorectal cancer, CRC, MFAP2 expression downstream of TWIST1 led not only to platinum resistance but also to a CSC phenotype [94]. TWIST1 induced a CSC phenotype in CRC cells that was responsible for irinotecan resistance [124]. Circulating breast cancer cells shared EMT and CSC markers and were resistant to eribulin mesylate [125]. However, as mentioned above, ovarian cancer stem cells are chemoresistant, but unlike most tumors, the CD44+ CSC population is decidedly epithelial in nature [120]. This finding underscores the need to consider each cancer and subtype as a unique disease and avoid generalizations in EMT effects.

**Table 1 ijms-24-17539-t001:** Summary of drugs whose efficacy is impacted by EMT. Applicable cancers, processes, or genes are involved, and the corresponding references are given. Asterisks denote cancers in which EMT has been correlated with sensitivity to the indicated drug as opposed to resistance.

Drug	Cancers	Genes/Pathways Impacted	References
Cisplatin	Ovarian *, mesothelioma, gastric, NSCLC, breast	Akt, miRNA, Rab31, eIF5A2, CDK1	[85,86,88,89,90,93,97,98,103]
Oxaliplatin	CRC	Akt, MFAP2, HK2, FHL3, Slug	[87,94,104,107]
5-Fluorouracil	CRC	Akt	[87]
Paclitaxel	Breast, ovarian	Akt2, CDK1	[86,91,103]
Resveratrol	Ovarian	Akt, hedgehog	[90]
Pterostilbene	Ovarian	STAT3	[91]
Doxorubicin	HCC	eIF5A2	[96]
Decitabine	AML	OGT	[105]
Pazopanib	Renal	Vascular mimicry, androgen receptor	[112]
Apatinib	Gastric	VEGFR2, COL1A2	[114]
Erlotinib	NSCLC	EGFR, Akt, MAPK	[115]
Gefitinib	NSCLC	EGFR, Akt, non-coding RNA, microRNA	[116,117]
Erastin	Gastric	CPEB1, ferroptosis	[118]
Olaparib	Ovarian *	HR	[119]
Nu-7441	Ovarian *	HR	[119]
ABT-888	Breast *	HR	[121]
Ophiobolin A	Breast *	EMT	[122]
Milademetan	Intimal sarcoma *	MDM2/p53	[123]
Irenotecan	CRC	CSC	[124]
Eribulin mesylate	Breast	Circulating tumor cells, CSCs	[125]
Multiple	HCC, CRC, gastric	MDR1	[108,109,110]

## 5. TWIST1 and EMT as Therapeutic Targets

### 5.1. Overview

Given the diverse roles being played by TWIST1 and other EMT factors across multiple tumor types, it is no surprise that targeting EMT has gained interest as a therapeutic strategy. According to Takeichi et al., one method of preventing metastasis is to inhibit EMT [126]. By the same token, EMT inhibition may reverse drug resistance and reduce the recurrence of tumors by blocking stem cell activity. As reviewed by Mirzaei et al., doxorubicin can induce EMT, which, in turn, confers resistance, making EMT an attractive target [127]. Targeting TWIST1 in particular was already being explored ten years ago, as reviewed by Khan et al. [128]. Similarly, a recent review has described multiple modalities for drugging vimentin [129]. A variety of approaches to anti-EMT therapy have been proposed, including the discovery and development of drugs against EMT or the use of gene knockdown and nanoparticle formulations. In some cases, existing drugs have been found to affect EMT, and these may also represent potential new cancer therapies. This section will describe examples of each of these strategies, and a summary is given in Table 2.

### 5.2. Drugs Targeting EMT

Natural products are an attractive source of novel anti-cancer compounds. Several have been shown to have activity not only against cancer cell proliferation but also against EMT. COM33, a derivative of the plant alkaloid 8-aminoisocorydine, was able to inhibit EMT induced by carboplatin treatment in ovarian cancer, reducing migration, invasion, and carboplatin resistance and synergizing with carboplatin in in vivo testing in the process [130]. Carboplatin treatment was also associated with increased ERK activity, which was prevented by COM33 administration. Furthermore, Peng and colleagues found that aloe emodin could reverse acquired gefitinib resistance in NSCLC as part of their study referenced earlier [116]. Nisin, a preservative produced as part of bacterial fermentation, was able to reduce TWIST1 expression in liver cancer, following the binding of nisin to the frizzled receptor FZD7 [131]. α-Linolenic acid reduced TWIST1 expression levels, increased its turnover, and led to EMT reversal in cell line models of TNBC [132]. Ophiobolin A, the fungus-derived compound introduced above, could prevent EMT and reduce the growth of CSC-rich tumors in an in vivo model of breast cancer [122].

The harmala alkaloid harmine and its derivatives have also been studied in this context, particularly as TWIST1 inhibitors. He et al. showed that harmine treatment could reduce breast cancer cell invasiveness through the inhibition of EMT [133], while Nafie and colleagues showed a similar effect, specifically through protein turnover of TWIST1 [134]. Harmine was also used to inhibit TWIST1 in luminal breast cancers, where it was able to resensitize palbociclib-resistant MCF7 cells to CDK4/6 inhibitors [135]. In 2021, Zhao et al. designed novel TWIST1 inhibitors using harmine as a base, and were able to improve on the IC50 of harmine when treating NSCLC cells [136]. Lu et al. took a similar approach the following year, although their focus was on harmine’s ability to intercalate DNA and thus synergize it with histone deacetylase inhibitors [137]. In 2023, Zhang et al. showed that another harmine derivative effectively blocked TNBC growth and metastasis in mice, including the regulation of CSCs [138]. A more extensive review of natural products capable of regulating CSCs was recently published by Reisenauer et al. [139]. Incidentally, harmine was also used to reduce TWIST1 activity in the model of ulcerative colitis described in the previous section [53].

### 5.3. Gene Knockdown and Nanoparticle Approaches

Transcription factors are often considered difficult to drug given their nuclear localization [140]. Even harmine, which appears to have promising anti-cancer effects in vitro, would be difficult to translate clinically given its toxicity and psychoactive properties [141,142]. Gene knockdown is an attractive alternative that shows some promise in treating difficult targets. In a 2023 study, Dogra et al. showed that the knockdown of doublecortin-like kinase 1 (DCLK1) reversed TGFβ-related EMT, including a reduction in vimentin, Snail, and Zeb2 but not TWIST1 [143]. However, their in vivo work was performed with a small molecule inhibitor of DCLK1, which reduced metastatic growth in combination with cisplatin. Furthermore, Bahar et al. showed that TWIST1 knockdown in platinum-resistant ovarian cancer cells sensitized them to a combination of cisplatin and the PARP inhibitor niraparib [144]. We and our colleagues have published extensively on the use of TWIST1 siRNA delivered by nanoparticles to combat cancer. To summarize, we showed that these nanoparticles could deliver payloads to breast tumors [145], reduce the growth of melanoma xenografts [146], and sensitize ovarian cancer cells to cisplatin [147]. A second generation of nanoparticles conjugated to hyaluronic acid further improved efficacy in ovarian cancer [148]. Nanoparticle formulations need not include siRNA to impact EMT; however. Dhanwal et al. demonstrated that hybrid gold–organic nanoparticles with a payload of the drug BZ6 reduced EMT markers, such as TWIST1, and prevented migration in in vitro studies of both breast and pancreatic cancer. Furthermore, a remarkable reduction in 4T1 tumor growth was seen following treatment in an in vivo model [149]. It remains to be seen what future developments will be possible with nanoparticle platforms such as these.

### 5.4. Existing Drugs with Effects against EMT

Recent work has identified existing drugs with effects on cancer EMT that could be repurposed to fight aggressive tumors. Harkening back to the links between EMT and fibrosis, pirfenidone, a pulmonary fibrosis drug, was shown to inhibit TGFβ and Smad2 activation in triple-negative breast cancer. Proliferation, motility, and EMT were reduced, and pirfenidone synergized with paclitaxel; however, millimolar doses were necessary [150]. Meanwhile, metformin, a mainstay of therapy for type II diabetes, inhibited TGFβ, EMT, and ERK/MAPK signaling in endometrial carcinoma driven by PTEN loss [151]. Furthermore, Hseu et al. examined the effects of antrodia salmonea fungal extract (AS), a common Taiwanese traditional medicine, on head and neck cancer. They found that AS inhibited HIF1α, TWIST1, and N-cadherin expression and reduced glycolytic flux, suggesting a link between EMT inhibition and a reversal of the Warburg effect [152]. In 2023, Ferraro et al. found that the β2 agonist formoterol, commonly used to treat chronic obstructive pulmonary disease, could oppose the effects of cigarette smoke extract on lung cancer cells. In particular, reactive oxygen species (ROS) and EMT were reduced by formoterol, suggesting that this agent may be useful in a lung cancer setting [153]. Meanwhile, in a study of cervical cancer, Mozafari et al. showed that the antiemetic aprepitant could reverse substance P-mediated MMP production and cellular migration, both of which are associated with EMT [33,154]. The same laboratory later found that aprepitant may have a role in blocking ROS in glioblastoma and breast and prostate cancers [155,156,157].

Cancer drugs may also be candidates for anti-EMT therapy. In a different kind of nanoparticle study, Jiang et al. found that encapsulating paclitaxel in nanoparticles made with biguanide-modified albumin rather than the unmodified formulation led to the inhibition of EMT and CSC properties in breast cancer [158]. Compared to albumin alone, the modified vehicles resulted in a greater reduction in vimentin expression, increased E-cadherin levels, decreased mammosphere formation, and fewer CSCs, as marked by CD44+/CD24− cell number. In another study, Du et al. found that the histone deacetylase inhibitor entinostat reduced the growth and migration of gastric cancer cells by targeting SALL4 and reversing the cadherin switch [159].

Thus, EMT and its effectors are responsible for altered drug responses and may also represent new therapeutic targets (Figure 2).

**Table 2 ijms-24-17539-t002:** Summary of drugs shown to inhibit EMT. For each drug, relevant cancers and cellular processes and targets are given, along with the corresponding references.

Drug/Approach	Cancers	Pathways Impacted	References
COM33	Ovarian	Migration, invasion, resistance	[130]
Nisin	Liver	Frizzled	[131]
α-Linolenic acid	Breast	TWIST1 turnover	[130]
Ophiobin A	Breast	CSCs	[122]
Harmine and derivatives	Breast, NSCLC (ulcerative colitis)	Invasion, TWIST1 turnover, DNA intercalation, CSCs	[53,133,134,135,136,137,138]
Gene knockdown	Ovarian, breast, melanoma	DCLK1, TWIST1	[143,144,145,146,147,148]
BZ6 nanoparticles	Breast, pancreatic	Migration, TWIST1	[149]
Pirfenidone	Breast	TGFβ, Smad2	[150]
Metformin	Endometrial	TGFβ, ERK/MAPK	[151]
Antrodia salmonea	Head and neck	HIF1α, glycolysis	[152]
Formoterol	Lung	Reactive oxygen species (ROS)	[153]
Aprepitant	Cervical, glioblastoma, breast, prostate	ROS, NFκB	[154,155,156,157]
Paclitaxel nanoparticles	Breast	EMT and CSCs, mammospheres	[158]
Entinostat	Gastric	Histone deacetylase, SALL4	[159]

## 6. Conclusions

The vast majority of cancer deaths are the result of metastatic disease, which is often recurrent and drug-resistant. EMT lies at the intersection of all of these phenomena, and this fact alone justifies the extensive research that is being performed in this area. The strongest indication that EMT plays a part in cancer development is that some EMT regulators can help tumors grow and spread [39,160]. Otherwise, benign tumor cells that do not invade or metastasize can get into the tissue around them and spread to faraway places because EMT occurs while the tumor grows [161]. The pathological phases of tumor progression support this model. Our present understanding of the transcriptional regulatory network underlying EMT is based on many studies over the past twenty years that have focused on EMT in this setting. Another indication of EMT complexity is that cells going through it are sometimes seen to have markers of both mesenchymal and epithelial cells, referred to as a hybrid phenotype. What aspects of EMT are sufficient for tumor spread, and whether hybrid cells are the migratory population, remains to be determined. However, independent of migration and invasion, whole new areas of EMT research have also opened.

In addition to significantly supporting tumor invasion and metastasis [162,163], EMT has been shown to imbue cancer cells with traits that make them harder to kill, i.e., stemness, resistance to chemotherapy, and the ability to evade the immune system [164]. These facts strongly support an increased focus on EMT in future work. Cancers such as ovarian and colorectal, which are known for frequent recurrence and metastasis, stand to benefit from a greater understanding of tumor cell stemness and tumor regrowth. We and colleagues recently reviewed the challenge presented by heterogeneous ovarian tumors, including stem cell populations [60].

Stemness and drug resistance are related phenomena since tumors often regrow from cells that survived initial rounds of therapy. TWIST1 and EMT have been tied to acquired resistance to a variety of agents, from traditional cytotoxic chemotherapy to targeted therapies (Table 1). Several pathways have been implicated in this effect across multiple tumor types, including survival signaling via PI3K and Akt and multi-drug resistance via MDR1/*ABCB1*. Moreover, treatment can drive EMT, either directly or through the selection of resistant subpopulations that are more mesenchymal. Taken together, these studies demonstrate that TWIST1 and EMT represent the hub of several barriers to successful therapy. We and others expect that the development of new treatment strategies targeting EMT will overcome these barriers and ultimately improve patient survival. However, some caution is needed. Recent studies in breast and ovarian cancer have shown that EMT can, in certain circumstances, be a marker of drug sensitivity rather than resistance. As we and our colleagues recently observed, in ovarian cancer this may be a case of stemness and EMT working in opposite directions, with both very epithelial and very mesenchymal cells exhibiting drug resistance [119] (Figure 3). Furthermore, how mesenchymal markers correspond to therapeutic efficacy in sarcomas remains to be seen. These cancers are derived from mesenchymal tissues, and, therefore, the concept of EMT may be moot. Nevertheless, the finding that TWIST1 was associated with favorable drug response in intimal sarcoma is cause for further work in this area [123].

Successful efforts to drug EMT have been a long time coming. For one thing, EMT is largely a transcriptional program, meaning would-be protein targets are transcription factors. Transcription factors tend to be confined to the nucleus, whose envelope represents an extra obstacle in drug delivery. Moreover, many transcription factors share homology, making finding specific drug target pairs difficult. TWIST1 does have a unique C-terminal domain termed the Twist Box or WR domain that mediates some of its binding interactions [33,58,102]. We and colleagues previously showed that this domain could be a drug target, but to date, no drugs have been developed [58]. One workaround for the challenge of targeting EMT is the use of RNA interference to knockdown EMT drivers. Recently, harmine and other natural products have been shown to affect EMT, including TWIST1 expression and processing [133,134,165,166]. Several groups have sought to improve the efficacy of harmine by creating analogs with better drug properties [136,137,138], and the further development of these and other similar compounds may be another avenue for the successful translation of an anti-EMT drug from the bench to the clinic.

Future work may also involve the use of protein degraders, such as proteolysis targeting chimeras (PROTACs), a newer therapeutic modality that is gaining traction. Rather than trying to develop small molecules to inhibit EMT transcription factor function or binding, it may be possible to direct the degradation of the target protein at the post-translational level. The challenge of target specificity among transcription factors is somewhat lessened by the recruitment of specific ubiquitin ligases, and the PROTAC approach has been applied to other targets previously considered “undruggable” [167]. Whether this approach will be used and whether it will be successful remain to be seen.

While the development of novel therapeutic strategies targeting tumor progression is already underway, we still have much to learn about the fundamental biology at work. Leveraging “big data” analyses of EMT gene expression, metastasis, recurrence, stemness, angiogenesis, and the modulation of drug response will likely be required to truly comprehend how all of these phenomena are connected at the molecular level. Additional new techniques that take context and cell/tissue type into account may also be useful. For instance, in the future, real-time EMT studies may be achievable due to the ability to envisage the morphology and mobility of singular tumor cells in a living organism [168].

In summary, further elucidation of EMT pathways will lead to a greater understanding of how tumors progress and become more deadly, and, in turn, lead to the development of novel therapeutics to address multiple challenges to successful cancer treatment simultaneously.

## Figures and Tables

**Figure 1 ijms-24-17539-f001:**
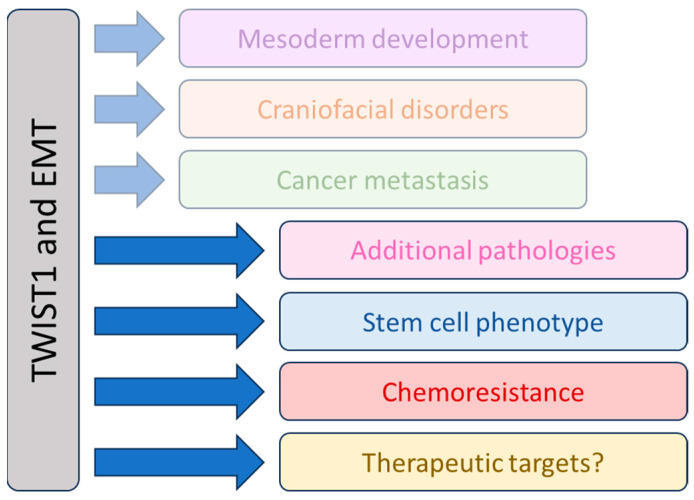
Overview of EMT and TWIST1 functions. TWIST1 and the EMT program are required for normal human development, and mutation or loss can lead to developmental syndromes. EMT is activated in cancers, and in addition to migration and metastasis, can also regulate other aspects of tumor progression. Established roles for EMT are shown as faded, while the more recent aspects of EMT, which will be the subject of our review, appear brightly.

**Figure 2 ijms-24-17539-f002:**
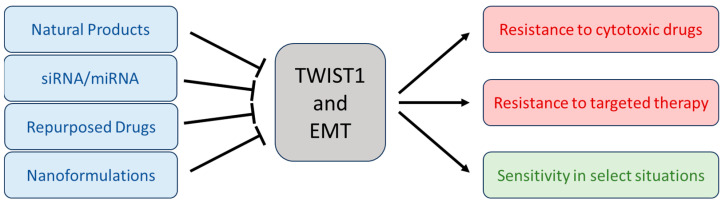
Summary of TWIST1, EMT, and cancer therapy. A variety of novel and repurposed therapies targeting TWIST1 and other EMT factors may help alleviate resistance to many current therapies. However, care should be taken to avoid inhibiting EMT in those rare situations where this process actually renders cells sensitive to treatment.

**Figure 3 ijms-24-17539-f003:**
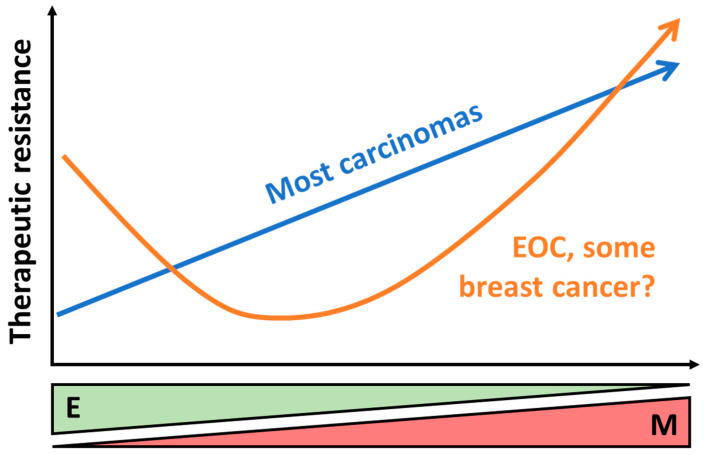
Relationship between EMT and drug responses. As EMT occurs, increased mesenchymal character (M) and loss of epithelial traits (E) often corresponds with increased resistance to a variety of drug classes (blue line). Ovarian cancer exhibits a different response profile, with epithelial, chemoresistant stem cells. There is some evidence that certain breast tumors share the characteristic of drug-sensitive mesenchymal cells, but how closely related these two cancers are remains unclear (orange line).

## Data Availability

No new data were created or analyzed in this study. Data sharing is not applicable to this article.

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
