# Peer review of "Shake It Up Baby Now: The Changing Focus on TWIST1 and Epithelial to Mesenchymal Transition in Cancer and Other Diseases"

_ijms, 2023, doi:10.3390/ijms242417539_

Round 1

Reviewer 1 Report

Comments and Suggestions for Authors

Needs to be revised.

Comments on the Quality of English Language

Require thorough proofreading by a native English speaker.

Reviewer 2 Report

Comments and Suggestions for Authors

Mirjat & Kashif et al. summarized the role of TWIST1 and the epithelial-to-mesenchymal transition (EMT) it regulated within cancer biology. They went through tumor progression, cancer stem cells and therapeutic responses, discussed how TWIST1 and EMT are involved in the above processes and how those findings can be incorporated into the treatment plan in the future.   

1.       The overall quality of the manuscript can be further improved by proofreading done by professional academic writers. The flow between the sentences is not as ideal as a review article.

2.       For all the drugs used in chemotherapy, it would be easier to get an overlook through a summarized form, with drug/chemical names, related cancers, involved pathways, and references. Including 2.2, 2.5, 2.6 and 5.2.

3.       Chapter 2 should be restructured, explain how TWIST1 regulated EMT protein expression and related signaling pathways first, then mention its relation with different types of drugs. It would be even better if you moved all the drug-related content from there to Chapter 5.

4.       In Chapter 2.5, I think you should mention anti-vascularization and EGFR-specific drugs in two separate chapters.

5.       Chapter 5, Are there any studies that showed regulation on EMT genes or proteins through drugs that are not majorly targeting EMT? I think that should be a short paragraph within Chapter 5.

Comments on the Quality of English Language

See above.

Reviewer 3 Report

Comments and Suggestions for Authors

17 November 2023

Ms. Ref. No.: ijms-2734011

Journal: International Journal of Molecular Sciences.

Title: Shake it Up Baby Now: the changing focus on TWIST1 and epithelial to mesenchymal transition in cancer and other diseases

Comments:

Thank you for your efforts in writing this review on a very pertinent topic. I have some observations where mentioned in the following paragraphs that will be useful for its improvement:

1-      The whole members of “TWIST1” in title, abstract and key words (such as lines 9 &23) are in capital but the term “Twist” in line 14 is not, is it right?

2-      There are many valuable articles in reference section of this manuscript that were cited, what is the inclusion and exclusion criteria of selecting these references?

3-      Rang of time in this article references were between 1983 and 2023, particularly the reference 3 which is about 1894, hence the question is about, why or how introduce this period? What was the main criteria for determining this rang? (Additionally , many of those are under 2010)

4-      Additionally, some of the Following reference can be included in the introduction part for more readability:   

·         https://doi.org/10.3390/ijms242216294

·         https://doi.org/10.1007/s00210-023-02551-0

·         https://doi.org/10.3390/ijms242015413  

·         https://doi.org/10.3390/ijms242216088

·         https://doi.org/10.1155/2022/8540403

·         https://doi.org/10.3390/ijms241612570  

·         https://doi.org/10.1007/s11033-021-06928-3

·         https://doi.org/10.1007/s12013-023-01171-y

5-      Moreover, adding suitable tables that summarize the text seems to be useful. For example the table that clarify the total articles that were studied and the numbers of excluded and so on.

6-      Finally, in sections 2.1 to 2.6 there are the association of EMT with other subjects and in section 5. TWIST1 and EMT as therapeutic targets, it seems suitable that summarizing these sections via tables.

Round 2

Reviewer 2 Report

Comments and Suggestions for Authors

The quality of the manuscript has improved a lot since the last revision, especially in the structure and writing.

Comments on the Quality of English Language

The flow between sentences and chapters has been improved after proofreading.